# Familiarity-Aware Evidence Compression for Retrieval-Augmented Generation

## Abstract

Retrieval-augmented generation (RAG) improves large language models (LMs) by incorporating non-parametric knowledge through evidence retrieved from external sources. However, it often struggles to cope with inconsistent and irrelevant information that can distract the LM from its tasks, especially when multiple evidence pieces are required. While compressing the retrieved evidence with a compression model aims to address this issue, the compressed evidence may still be unfamiliar to the target model used for downstream tasks, potentially failing to utilize the evidence effectively. We propose FAVICOMP (**FA**miliarity-aware **EVI**dence **COMP**ression), a novel training-free evidence compression technique that makes retrieved evidence more familiar to the target model, while seamlessly integrating parametric knowledge from the model. Specifically, FAVICOMP proactively composes the compressed evidence in a way to lower the perplexity of the target model by combining decoding probabilities from both the compression model and the target model to generate context that is more familiar to the target model. This approach balances the integration of parametric and non-parametric knowledge, which is especially helpful in complex tasks where the retrieved evidence set may not contain all the necessary information. Experimental results show that FAVICOMP consistently outperforms most recent evidence compression baselines across multiple open-domain QA datasets, improving accuracy by up to 23.91% while achieving high compression rates. Additionally, we demonstrate the effective integration of both parametric and non-parametric knowledge during evidence compression.

## 1 Introduction

Retrieval-augmented generation (RAG) has become a common paradigm for large language models (LMs) to leverage external knowledge beyond their inherent knowledge boundaries to perform better in knowledge-intensive tasks such as open-domain question answering (QA) (Lewis et al., 2020; Izacard & Grave, 2021; Guu et al., 2020) and fact-checking (Pan et al., 2023; Li et al., 2024c). In particular, incorporating multiple evidence pieces is crucial in solving complicated tasks such as multi-hop and complex reasoning (Trivedi et al., 2023; Jiang et al., 2023b; Li et al., 2024b; Lu et al., 2023), which require various sources of information to solve the questions.

Nevertheless, RAG often struggles to cope with inconsistent and irrelevant information from the multiple evidence set, which can interfere with downstream tasks (Shi et al., 2023). This highlights the need for *compression-based* RAG (Jiang et al., 2023a; Xu et al., 2024; Yoon et al., 2024) to identify and retain only the essential information for the LMs to utilize effectively. Traditionally, compression-based RAG has focused on reranking documents or sentences by relevance and then incorporating a top-ranked subset (Nogueira et al., 2020; Zhuang et al., 2023; Wang et al., 2023c) or compressing the documents into an abstractive summary that retains only essential context (Jiang et al., 2023a; Xu et al., 2024; Yoon et al., 2024). However, the compressed evidence might be unfamiliar to the LM employed for the downstream task (referred to as the target model), particularly due to discrepancies in the pretrained internal knowledge and prompt preferences between the compression model and the target model (Gonen et al., 2023; Lee et al., 2024; Li et al., 2024a; Mallen et al., 2023). When LMs encounter unfamiliar contextual information, they often fail in balancing parametric and non-parametric knowledge, either by overly relying on their parametric knowledge

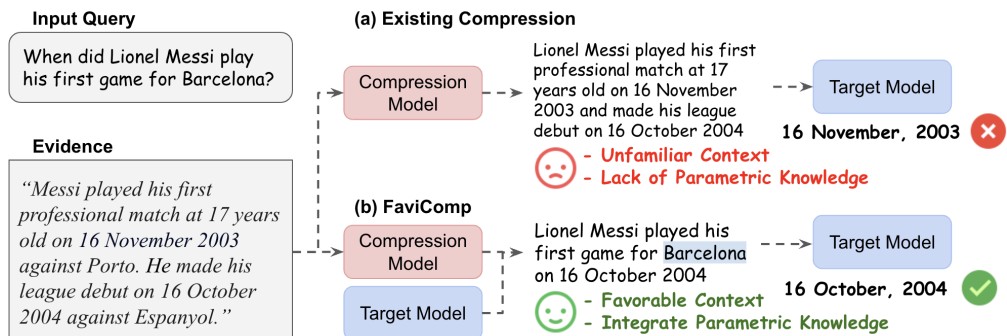

Figure 1: An overview of FAVICOMP. Instead of relying solely on compressed evidence from the compression model (upper), FAVICOMP familiarizes the compressed evidence to the target model while integrating parametric knowledge through ensemble decoding, resulting in improved downstream performance (lower).

(Longpre et al., 2021; Wang et al., 2023a; Zhou et al., 2023) or by utilizing retrieved evidence without considering its relevance to the input (Wu et al., 2024).

To address these challenges, we propose FAVICOMP (**FA**miliarity-aware **EVI**dence **COMP**ression), a training-free evidence compression method that makes retrieved multi-evidence more familiar to the target model, while seamlessly integrating parametric knowledge from the model. Inspired by the prior findings that an LM's familiarity with a prompt is generally reflected by low perplexity (Liu et al., 2024; Gonen et al., 2023; Wang et al., 2023b), FAVICOMP proactively composes the compressed evidence in a way to lower the perplexity of the target model. Specifically, FAVICOMP leverages the decoding probabilities of two LMs, a *compression model* and the *target model*. The compression model is instructed to summarize the raw evidential documents into a relevant context to the input, while the target model is instructed to generate relevant context without referencing the documents. Instead of directly selecting the highest probability token from the compression model at each decoding step, we ensemble the token logits from both the compression and target models and then select the token with the highest probability from this combined set. This ensemble decoding therefore constrains the token search space of the compression model to those with lower perplexity for the target model, making the context more familiar to the target model (Liu et al., 2024).

Furthermore, FAVICOMP potentially synergizes the retrieved knowledge with the target model's parametric knowledge introduced during ensemble decoding. FAVICOMP can effectively discern when to leverage internal or external knowledge, which is particularly beneficial in the presence of noisy contextual evidence in complex tasks such as multi-document or multi-hop QA (Wang et al., 2024).

FAVICOMP brings along key advantages of RAG for complex tasks from two perspectives. On the one hand, it is capable of compressing multiple augmented documents to a more favorable form to the target model. This mechanism not only helps the model better comprehend the essential evidence in the retrieval augmentation but also better balances knowledge utility in both the evidential context and the model's parametric memory. On the other hand, it is a training-free and model-agnostic approach that can be easily plugged into any RAG processes

Our experiments show that FAVICOMP outperforms most recent evidence compression baselines in five open-domain QA datasets, improving accuracy by up to 23.91% while maintaining high compression rates. Additionally, we conduct ablation studies by varying the degree of decoding ensemble and analyzing its impact on performance and context perplexity. Moreover, we investigate how FAVICOMP effectively integrates parametric and non-parametric knowledge during evidence compression.

## 2 METHOD

We present FAVICOMP, a decoding-time evidence compression method that familiarizes retrieved evidence with the target model while synergizing them with the model's parametric knowledge. We

first illustrate the motivation for FAVICOMP in §2.1 and provide the preliminaries of compression-based RAG in §2.2, followed by a detailed definition of our proposed framework in §2.3.

## 2.1 MOTIVATION AND METHOD OVERVIEW

Standard RAG faces the challenge of LMs struggling to address inconsistent and irrelevant information from multiple evidence pieces, which can interfere with downstream tasks (Shi et al., 2023). Previous research has primarily concentrated on question-focused compression (Jiang et al., 2023a; Xu et al., 2024; Yoon et al., 2024); however, this approach may lead to suboptimal performance in downstream tasks due to the compressed evidence's potential unfamiliarity with the target model employed. This unfamiliarity arises from discrepancies in pretrained internal knowledge and prompt preferences between the compression model and the target model (Gonen et al., 2023; Lee et al., 2024; Mallen et al., 2023). Furthermore, the unfamiliarity often leads to failure in balancing parametric and non-parametric knowledge, either by overly relying on their parametric knowledge (Longpre et al., 2021; Wang et al., 2023a; Zhou et al., 2023) or by using retrieved evidence without considering its relevance to the input (Wu et al., 2024). To address this issue, FAVICOMP introduces a novel approach that compresses evidence that better aligns with the target model's preferences while seamlessly integrating parametric knowledge into the compressed evidence using a novel ensemble decoding technique, thereby improving its performance on downstream tasks.

Fig. 1 illustrates the overview of FAVICOMP. In this example, FAVICOMP makes the compressed evidence more favorable to the target model and leverages its parametric knowledge to supplement the missing evidence (*"Lionel Messi made his league debut in Barcelona"*), effectively combining evidential and parametric knowledge.

## 2.2 COMPRESSION-BASED RETRIEVAL AUGMENTED GENERATION

Given a set of $k$ retrieved evidence snippets $D = \{d_1, d_2, \ldots, d_k\}$ and a textual input sequence $x$, standard RAG aims to generate an output sequence $y$, conditioned on both $D$ and $x$. However, standard RAG directly utilizes $D$ which often contains irrelevant information to $x$, potentially confusing the target model in downstream tasks (Shi et al., 2023). Thus, the compression-based RAG uses an additional compression model to condense $D$ into a concise and input-relevant context $c$, which is then used in place of $D$ during the downstream generation process. Thus, the compression-based RAG is formalized as:

$$y^* = \arg\max_y P_t(y \mid x, \hat{c}),$$

$$\hat{c} = P_c(c \mid x, [d_1, d_2, \ldots, d_k]),$$

where $y^*$ is the final output sequence, $[\cdot, \cdot]$ denotes concatenation, and $P_t$ and $P_c$ represent the probability distributions of the target and compression models, respectively. In this work, we consider any natural language prompting tasks, such as open-domain QA tasks, where $x$ represents the input prompt (also known as the query in QA tasks) and $y^*$ denotes the output sequence.

The compression model's objective is to produce a concise yet informative summary $c$ of the evidential documents $D$ that captures the essential information relevant to the input query $x$. We use an unsupervised approach, where the model is instructed to generate a query-relevant summary of $D$ in a zero-shot manner using an evidence compression instruction prompt, denoted as $I_{comp}$, such as the one below:

> **Evidence Compression Instruction**
>
> Given a question and multiple document snippets, generate one summarized context that is helpful to answer the question.

Specifically, the evidence compression is done in an auto-regressive way formalized as,

$$P_c(c \mid I_{comp}, x, D) = \prod_{i=1}^{|c|} P_c(c_i \mid I_{comp}, x, D, c_{<i}),$$

where $|c|$ is the length of the summary $c$.

## 2.3 ENSEMBLE DECODING FOR FAVICOMP

Simple compression techniques might lead to subpar performance in downstream tasks because the compressed evidence may not be familiar to the target model. To better align the context to the target model, FAVICOMP proactively composes it to lower the target model's perplexity by introducing a constraint in decoding space from the target model during the evidence compression. FAVICOMP achieves this goal through ensemble decoding, which involves a multiplicative ensemble of two LMs—compression model and target model—at each decoding step.

Specifically, the target model is directed to generate a context $c$ that would be helpful in answering the question $x$ without referencing the evidence set. This is also done in zero-shot using a context generation instruction prompt $I_{gen}$ such as:

> **Context Generation Instruction**
>
> Given a question, generate a context that is helpful to answer the question.

The context generation is also performed in an auto-regressive fashion, represented as:

$$P_t(c \mid I_{gen}, x) = \prod_{i=1}^{|c|} P_t(c_i | I_{gen}, x, c_{<i}),$$

where $|c|$ denotes the length of the generated context $c$.

Once the compression model and the target model generate their respective probability distributions for the next token, the subsequent token is chosen by maximizing the weighted sum of the log probabilities from both models. The selected token is the continuation of the previously generated text aligned with their objectives. This process is formalized as follows:

$$c_i = \arg\max_{c_i \in V}((1 - \alpha) \cdot \log P_c(c_i | I_{comp}, x, D, c_{<i}) + \alpha \cdot \log P_t(c_i | I_{gen}, x, c_{<i})),$$

where $c_i$ is the subsequent token, and $\alpha$ is the ensemble coefficient that weighs between the two probability distributions. We demonstrate how the coefficient $\alpha$ impacts both the perplexity and the downstream performance in §4.2.

Ensemble decoding proactively shifts the token search space in evidence compression by upweighting those tokens with lower perplexity from the target model's perspective (Liu et al., 2024), resulting in a compressed evidence that is more familiar to the target model. Note that since both objectives ultimately share the goal of generating context relevant to the question, combining the logits ensures alignment with this ultimate goal.

In addition, ensemble decoding enables FAVICOMP to seamlessly integrate both retrieval knowledge from the external evidence set and the target model's parametric knowledge. Specifically, FAVICOMP selects the $\arg\max$ token from the target model only when the token's probability is higher than that of the compression model, demonstrating that FAVICOMP draws on parametric knowledge only when necessary—potentially when the compression model is uncertain about the next token. This is particularly beneficial for complex tasks like multi-document QA, where the evidence set may not include all the necessary information (Mallen et al., 2023). In such cases, the missing information in compressed evidence can be supplemented by tokens generated from context generation by the target model, which is entirely based on parametric knowledge. We demonstrate in §4.3 and §5 that FAVICOMP can incorporate knowledge from both sources effectively, leading to a performance boost compared to compression methods that solely focus on distilling knowledge from the evidence set.

## 3 EXPERIMENTAL SETTINGS

We assess the effectiveness of FAVICOMP on knowledge-intensive QA tasks. In this section, we delve into the details of the experimental settings.

### 3.1 DATASETS

We evaluate FAVICOMP on five open-domain QA datasets, including two single-document QA datasets, Natural Questions (NQ) (Kwiatkowski et al., 2019) and TriviaQA (TQA; Joshi et al. 2017), and three multi-document QA datasets, HotpotQA (Yang et al., 2018), 2WikiMultiHopQA (2WikiMQA; Ho et al. 2020), and MuSiQue (Trivedi et al., 2022). Following prior studies (Asai et al., 2023; Xu et al., 2024), we evaluate the performance on the development set of each dataset and use three evaluation metrics, i.e. Accuracy (Acc), token-level F1 and compression rate (Comp) which is calculated as $\frac{\text{\# of tokens in retrieved documents}}{\text{\# of tokens in compressed documents}}$.

### 3.2 IMPLEMENTATION DETAILS

For all the comparison methods, we utilize three LMs as the target model to tackle downstream QA tasks with RAG, i.e. `Llama3-8B-Instruct`[1], `Mistral-7B-Instruct`[2] and `Mixtral-8x7B-Instruct`[3]. For each question, we retrieve five documents from 2018 Wikipedia corpus (Karpukhin et al., 2020) using Contriever-MSMARCO[4] (Izacard et al., 2021), so as to be consistent with previous studies (Xu et al., 2024; Yoon et al., 2024).

For FAVICOMP, we employ three compression and target model pairs: (1) `Llama3.2-3B-Instruct` and `Llama3-8B-Instruct` as the compression models and `Llama3-8B-Instruct` as the target model, (2) `Mistral-7B-Instruct` as the compression model and `Mixtral-8x7B-Instruct` as the target model, and (3) same `Mistral-7B-Instruct` as the compression model and the target model (Appx. §B.1). Also, we set $\alpha$ to 0.5 by default, for which more analyses are given in §4.2. The prompts used in the experiment are presented in Appx. §C.

### 3.3 BASELINES

We consider the following categories of baselines. (1) **No Context**: RAG without any context. (2) **Gold Compression**: RAG using directly relevant evidence from the retrieved documents if they exist. (3) **Raw Document**: RAG with raw documents that have not undergone any compression. (4) **Generated Context** (Yu et al., 2023): RAG with context generated by the same LM as the target model. This is equivalent to FAVICOMP with $\alpha = 1$, as we rely solely on the target model to generate context when $\alpha = 1$. (5) **Reranking-based Methods**: We rerank sentences in the evidence set and choose top-ranked sentences as the context. We utilize two rerankers—Sentence-BERT (Reimers & Gurevych, 2020) and RECOMP-extractive (Xu et al., 2024). (6) **Compression-based Methods**: We employ four compressors—LongLLMLingua (Jiang et al., 2023a), RECOMP-abstractive (Xu et al., 2024), CompAct (Yoon et al., 2024), and Zero-shot Summarization. Zero-shot Summarization is instructed to summarize the evidence set into a concise summary based on the question, using the same LM as the target model. This is equivalent to FAVICOMP with $\alpha = 0$, as we depend entirely on the compression model without any intervention from the target model. A detailed explanation of the implementation of the baselines is provided in Appx. §A.

## 4 EXPERIMENTAL RESULTS

In this section, we compare the overall performance of FAVICOMP with other baselines across the five datasets (§4.1), explore the impact of ensemble coefficient $\alpha$ on performance and perplexity (§4.2), investigate how effectively FAVICOMP incorporate parametric and non-parametric knowledge (§4.3), and compare the compression rates with other baselines (§4.4).

### 4.1 MAIN RESULTS

The overall performance of FAVICOMP and the baselines across the five datasets are presented in Tab. 1 and Tab. 3. To start with, the compression-based methods consistently outperform the

---

[1] https://huggingface.co/meta-llama/Meta-Llama-3-8B-Instruct

[2] https://huggingface.co/mistralai/Mistral-7B-Instruct-v0.3

[3] https://huggingface.co/mistralai/Mixtral-8x7B-Instruct-v0.1

[4] https://huggingface.co/facebook/contriever-msmarco

| Methods | Size | NQ | | | TQA | | | HotpotQA | | | 2WikiMQA | | | MuSiQue | | |
|---|---|---|---|---|---|---|---|---|---|---|---|---|---|---|---|---|
| | | Acc | F1 | Comp | Acc | F1 | Comp | Acc | F1 | Comp | Acc | F1 | Comp | Acc | F1 | Comp |
| *Llama3-8B-Instruct* | | | | | | | | | | | | | | | | |
| Gold Compression | - | - | - | - | - | - | - | 42.3 | 51.3 | - | 35.7 | 40.0 | - | 10.2 | 17.7 | - |
| No Context | - | 26.9 | 31.9 | - | 57.2 | 61.2 | - | 19.1 | 25.5 | - | 20.5 | 25.0 | - | 5.4 | 13.0 | - |
| Raw Document | - | 42.6 | **47.1** | - | 67.6 | 70.8 | - | 30.3 | 38.7 | - | 22.0 | 26.8 | - | 8.2 | 15.0 | - |
| Generated Context | - | 32.3 | 36.6 | - | 59.7 | 62.4 | - | 22.7 | 29.7 | - | 24.8 | 28.7 | - | 7.6 | 14.8 | - |
| Sentence-BERT | 110M | 30.3 | 35.4 | 21.13 | 59.2 | 62.9 | 20.61 | 22.4 | 29.6 | 10.30 | 18.1 | 22.9 | 9.96 | 7.7 | 14.8 | 10.18 |
| RECOMP-extractive | 110M† | 33.7 | 38.1 | 19.45 | 59.4 | 62.8 | 18.86 | 22.5 | 29.8 | 9.47 | 18.0 | 22.4 | 9.17 | 8.1 | 15.5 | 9.24 |
| LongLLMLingua | 7B† | 35.4 | 40.9 | 1.87 | 64.8 | 67.6 | 1.84 | 25.9 | 34.7 | 1.83 | 19.2 | 24.2 | 1.83 | 7.7 | 14.4 | 1.83 |
| RECOMP-abstractive | 775M† | 39.3 | 43.3 | 17.96 | 62.9 | 66.1 | 17.79 | 27.0 | 34.8 | 19.72 | 20.5 | 25.0 | 32.06 | 7.3 | 14.8 | 32.05 |
| CompAct | 7B† | 42.3 | 46.1 | 8.85 | 67.0 | 69.7 | 8.92 | 29.8 | 37.5 | 9.45 | 21.4 | 26.6 | 10.71 | 9.2 | 16.9 | 8.96 |
| Zero-shot Summarization | 3B | 39.4 | 43.2 | 14.12 | 64.2 | 67.1 | 17.12 | 30.1 | 38.5 | 18.75 | 25.7 | 31.1 | 21.39 | 7.7 | 15.3 | 16.19 |
| | 8B | 41.3 | 45.1 | 13.87 | 66.3 | 69.5 | 16.58 | 30.2 | 38.6 | 17.38 | 22.3 | 28.1 | 18.98 | 8.3 | 16.3 | 15.50 |
| **FAVICOMP** | 3B | **42.8** | 46.8 | 16.43 | 68.0 | 70.9 | 22.40 | **33.0** | **41.6** | 22.55 | **29.6** | **35.2** | 23.10 | 10.8 | 19.9 | 18.95 |
| | 8B | 42.3 | 46.6 | 15.79 | **68.4** | **71.5** | 20.99 | 32.3 | 41.0 | 21.49 | 27.6 | 33.6 | 22.41 | **11.4** | **20.1** | 19.06 |
| *Mixtral-8x7B-Instruct* | | | | | | | | | | | | | | | | |
| Gold Compression | - | - | - | - | - | - | - | 48.2 | 55.1 | - | 49.9 | 51.9 | - | 12.9 | 18.6 | - |
| No Context | - | 36.7 | 38.4 | - | 68.9 | 72.0 | - | 25.1 | 31.6 | - | 32.5 | 35.9 | - | 6.4 | 11.8 | - |
| Raw Document | - | **46.3** | 42.1 | - | 72.1 | 71.1 | - | 34.0 | 39.0 | - | 32.9 | 36.3 | - | 10.1 | 15.6 | - |
| Generated Context | - | 33.6 | 33.9 | - | 61.4 | 62.9 | - | 26.5 | 32.9 | - | 30.2 | 34.3 | - | 7.2 | 13.4 | - |
| Sentence-BERT | 110M | 36.8 | 36.8 | 21.13 | 67.0 | 68.7 | 20.61 | 28.3 | 34.5 | 10.13 | 32.5 | 36.2 | 9.76 | 9.9 | 15.2 | 10.07 |
| RECOMP-extractive | 110M† | 38.0 | 37.9 | 19.42 | 66.7 | 68.0 | 18.81 | 28.7 | 34.3 | 9.30 | 31.8 | 34.9 | 9.01 | 9.4 | 15.6 | 9.11 |
| LongLLMLingua | 7B† | 40.1 | 39.4 | 1.96 | 70.5 | 71.0 | 1.96 | 32.0 | 38.3 | 1.95 | 31.9 | 36.1 | 1.93 | 9.7 | 15.9 | 1.94 |
| RECOMP-abstractive | 770M† | 42.1 | 41.3 | 17.55 | 68.4 | 69.4 | 17.47 | 32.3 | 38.5 | 19.39 | 32.2 | 36.2 | 31.20 | 7.9 | 13.6 | 31.18 |
| CompAct | 7B† | 44.1 | 43.4 | 8.83 | 70.3 | 71.4 | 8.92 | 35.2 | 41.6 | 9.45 | 35.9 | 39.5 | 10.67 | 11.2 | 16.9 | 8.94 |
| Zero-shot Summarization | 7B | 42.1 | 40.6 | 8.65 | 65.9 | 67.0 | 10.43 | 31.4 | 38.1 | 11.71 | 28.5 | 32.8 | 14.35 | 8.4 | 13.8 | 10.26 |
| **FAVICOMP** | 7B | 43.6 | **44.5** | 7.30 | **72.6** | **73.9** | 8.21 | **36.3** | **44.4** | 8.89 | **40.5** | **45.2** | 10.26 | **13.4** | **19.9** | 8.42 |

Table 1: Experimental results on five open-domain QA datasets. **Size** column represents the size of the compression model used for each method. † indicates a fully-supervised compression model, where the reranker or the compressor is trained. For the experiment with `Llama3-8B-Instruct`, Zero-shot Summarization and FAVICOMP use `Llama3.2-3B-Instruct` and `Llama3-8B-Instruct` as the compression model, shown as 3B and 8B in the **Size** column. The best Accuracy and token-level F1 scores for each dataset are in bold.

reranking-based methods, due to the fact the reranking-based methods are prone to losing more question-relevant information by discarding lower-ranked sentences. Next, FAVICOMP outperforms all other baselines across all the datasets, except for the Gold Compression which is regarded as the upper bound of the performance. It is noteworthy that FAVICOMP, as a training-free, decoding-time strategy, outperforms supervised baselines even with the 3B parameters compression model. For the MuSiQue dataset, FAVICOMP even outperforms Gold Compression baseline which can be viewed as a perfect compressor. This demonstrates that explicitly incorporating parametric knowledge from the target model can significantly enhance performance in multi-document QA, even when the context is imperfect.

Moreover, it is surprising that most of the supervised compression-based methods are excelled by the Raw Document baseline. This indicates that existing methods are likely to fall short of retaining essential supportive information while compressing the evidence documents. Additionally, LongLLMLingua and RECOMP-abstractive perform worse than Zero-shot Summarization with similar or smaller size compression model. This may be possibly due to the use of smaller base model for compression (`T5-large` for RECOMP-abstractive), but it also suggests that knowledge distillation from larger teacher LM to the smaller compression model may not generalize well, as the context preferences and prior knowledge of the target model and the teacher model are likely to differ. We conduct a head-to-head experiment on RECOMP-abstractive by using the same base compression model as FAVICOMP for a more fair comparison in Appx. §B.2.

Furthermore, despite using the same base model for the compression model (`Mistral-7B-Instruct`), the training-free FAVICOMP outperforms CompAct, which trains the compression model using knowledge distillation to generate and evaluate summaries of retrieved documents. This also indicates that knowledge distilled from a teacher model may not always be effectively transferable to the target model due to discrepancies in context preference and prior knowledge. In contrast, the superior performance of FAVICOMP is attributed to its ability to

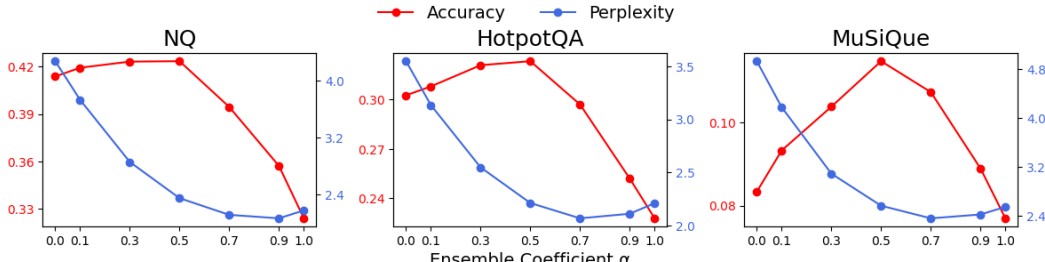

Figure 2: Impact of coefficient $\alpha$ on performance and perplexity.

familiarize evidence with the target model and its effective incorporation of parametric knowledge from ensemble decoding.

Finally, given that Zero-shot Summarization corresponds to FAVICOMP with $\alpha = 0$ and Generated Context corresponds to FAVICOMP with $\alpha = 1$, the fact that FAVICOMP outperforms both baselines highlights its ability to effectively incorporate tokens from both sources—evidence summary and generated context. This results in superior performance compared to relying on just one source alone.

## 4.2 IMPACT OF ENSEMBLE COEFFICIENT ON PERFORMANCE AND PERPLEXITY

Fig. 2 illustrates how performance and perplexity change as the ensemble coefficient $\alpha$ is varied across the values $\{0.0, 0.1, 0.3, 0.5, 0.7, 0.9, 1.0\}$ on NQ, HotpotQA and MuSiQue datasets. We calculate the perplexity of the compressed evidence conditioned on the preceding inputs, i.e. instruction, demonstrations, and the question. For all the datasets, performance is the highest when $\alpha = 0.5$, indicating that proactively lowering perplexity by equally weighting both input sources yields the best results. When $\alpha$ is below 0.5, performance improves as the perplexity of compressed evidence decreases, which aligns with the previous works (Liu et al., 2024; Gonen et al., 2023). However, when $\alpha$ exceeds 0.5, performance declines as perplexity decreases due to the lack of evidential knowledge during evidence compression. Additionally, when $\alpha$ reaches 0.9 or 1.0, there is a slight rise in the perplexity due to LM's increased uncertainty with limited evidential knowledge. Results for other datasets are included in Fig. 4.

## 4.3 INTEGRATION OF PARAMETRIC AND NON-PARAMETRIC KNOWLEDGE

The effective integration of parametric and non-parametric knowledge is crucial for complex tasks such as multi-document QA, where the evidence set may not contain all the necessary information. To this end, we evaluate how effectively FAVICOMP incorporates parametric knowledge from the target model and non-parametric knowledge from the compression model on the multi-document QA datasets. We begin by dividing the test samples of each dataset into evidence-relevant and evidence-irrelevant subsets, using the $Hits$ metric. The $Hits$ metric is set to 1 (evidence-relevant) if the retrieved evidence set contains the correct answer, and 0 (evidence-irrelevant) if it does not. We then assess the downstream performance of each subset. The underlying intuition is that if a method performs better on the evidence-relevant subset, it suggests that the method is more effectively utilizing the provided evidential knowledge. Conversely, if a method excels on the evidence-irrelevant subset, it indicates that the method is more effectively leveraging parametric knowledge without relying on potentially irrelevant evidence.

The left figure of Fig. 3 compares the accuracy in $Hits = 0$ and $Hits = 1$ subsets across the datasets. We compare FAVICOMP with the top-performing unsupervised compression method, Zero-shot Summarization, and the most competitive supervised compression method, CompAct. Compared to the other two baselines, FAVICOMP performs better in the $Hits = 0$ subset while performing comparably in the $Hits = 1$ subset. This proves that FAVICOMP effectively relies on parametric knowledge rather than evidential knowledge when faced with irrelevant evidence, while maintaining similar effectiveness in utilizing evidential knowledge when relevant evidence is present.

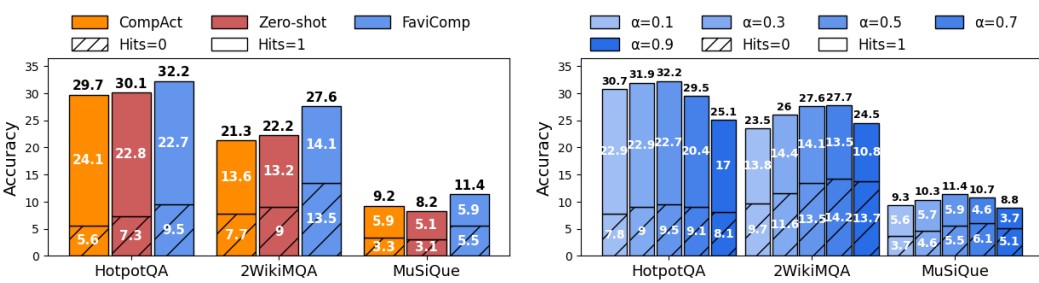

Figure 3: Accuracy of baselines (left) and FAVICOMP with various $\alpha$ values (right) on $Hits = 0$ and $Hits = 1$ subset of multi-document QA datasets.

Interestingly, even though CompAct generally performs better on the $Hits = 1$ subset compared to Zero-shot Summarization, it underperforms relative to Zero-shot Summarization on the $Hits = 0$ subset. This suggests that the training may have been biased towards utilizing solely evidential knowledge, rather than effectively leveraging both sources in synergy.

We also evaluate the performance of FAVICOMP with various $\alpha$ values under this setting. The right figure of Fig. 3 shows that $\alpha = 0.5$ or $\alpha = 0.7$ performs the best on the $Hits = 0$ subset, while performance declines as $\alpha$ deviates further from the value. This pattern in the $Hits = 0$ subset mirrors the overall performance trend, suggesting that appropriately utilizing parametric knowledge when the evidence is irrelevant is crucial to the overall performance. In the $Hits = 1$ subset, performance remains consistent for $\alpha$ values up to 0.5 but decreases significantly when $\alpha$ exceeds 0.5 due to the diminished utilization of the relevant evidential context.

## 4.4 COMPRESSION RATE COMPARISONS

Since one of the functionalities of compression-based RAG is to reduce the number of tokens from the evidence while keeping its essential information, we report the compression rate in Tab. 1. Overall, reranking-based methods, RECOMP-abstractive and FAVICOMP consistently score the highest compression rates. Reranking-based methods achieves high compression since they only select one or two sentences that may contain the answer to the question, but the information loss is more significant compared to other methods. RECOMP-abstractive exhibits high compression rates because the compression model is trained to output an empty string when no relevant evidence is found, which is often the case in multi-document QA datasets. FAVICOMP compresses the evidence to make it familiar to the target model by lowering its perplexity at each decoding step, typically resulting in a shorter context. Notably, when compared to Zero-shot Summarization, which is equivalent to FAVICOMP with $\alpha = 0$, FAVICOMP consistently achieves higher compression rates. This demonstrates that the ensemble decoding strategy, combining token logits from both evidence compression and context generation, leads to greater compression efficiency.

## 5 CASE STUDY

Tab. 2 presents two examples from HotpotQA to illustrate how FAVICOMP effectively familiarizes evidence while seamlessly integrating both parametric and non-parametric knowledge during evidence compression. We compare its output with Raw Document, which does not apply any compression, and Zero-shot Summarization, which is equivalent to FAVICOMP with $\alpha = 0$.

In both examples, Raw Document fails to produce the correct answer, even though the evidence contains the necessary information, highlighting the need for effective evidence compression. In the first example, while the difference between the compressed evidence from Zero-shot Summarization and FAVICOMP appears subtle, FAVICOMP delivers the correct answer with a lower perplexity in compression, underscoring the significance of evidence familiarization. The second example highlights the importance of parametric knowledge when the retrieved evidence set lacks complete information. Since the evidence set does not mention "Skeptic," Zero-shot Summarization intro-

| **Question**: This film is an adaption of a Jacques Offenbach's opera that was written by a Hungarian British screenwriter? | | | |
|---|---|---|---|
| **Methods** | **(Compressed) Evidence** | **Prediction** | **Perplexity** |
| Raw Document | ...(skip)... The Tales of Hoffmann is a 1951 British Technicolor film adaptation of Jacques Offenbach's opera "The Tales of Hoffmann", written, produced and directed by the team of Michael Powell and Emeric Pressburger working under the umbrella of their production company, The Archers. | Emeric Pressburger ✗ | 12.429 |
| Zero-shot Summarization | The 1951 film "The Tales of Hoffmann" is an adaptation of Jacques Offenbach's opera, written, produced, and directed by Michael Powell and Emeric Pressburger. | Emeric Pressburger ✗ | 2.298 |
| FAVICOMP | The 1951 film "The Tales of Hoffmann" is an adaptation of Jacques Offenbach's opera, written by Emeric Pressburger, a Hungarian-British screenwriter, and directed by Michael Powell and Emeric Pressburger. | The Tales of Hoffmann ✓ | 1.959 |

| **Question**: Which magazine was first published earlier, The Chronicle of Philanthropy or Skeptic? | | | |
|---|---|---|---|
| **Methods** | **(Compressed) Evidence** | **Prediction** | **Perplexity** |
| Raw Document | The Chronicle of Philanthropy is a magazine that covers the nonprofit world. ...(skip)... It was founded in 1988 by editor Phil Semas and then managing editor Stacy Palmer. ...(skip)... Philanthropy (magazine) Philanthropy is a quarterly magazine published by the Philanthropy Roundtable. First published as a newsletter in 1987, "Philanthropy" became a glossy magazine in 1996. | Philanthropy ✗ | 4.856 |
| Zero-shot Summarization | The Chronicle of Philanthropy was founded in 1988, while Philanthropy magazine was first published as a newsletter in 1987 and became a glossy magazine in 1996. | Philanthropy magazine ✗ | 3.196 |
| FAVICOMP | The Chronicle of Philanthropy was first published in 1988, while Skeptic was first published in 1992. | The Chronicle of Philanthropy ✓ | 1.345 |

Table 2: Case study of evidence compression: FAVICOMP vs. Raw Document and Zero-shot Summarization. For FAVICOMP, the colors red and blue highlight tokens that are the $\arg\max$ of the compression model and the target model, respectively. Purple indicates a token that is the $\arg\max$ of neither model. Tokens with no coloring represent those that are the $\arg\max$ of both models.

duces irrelevant information ("Philanthropy magazine"), ultimately leading to an incorrect answer. In contrast, FAVICOMP integrates parametric knowledge about "Skeptic" and incorporates it into the evidence compression. Notably, FAVICOMP selects the $\arg\max$ token from the target model only when the token's probability is higher than that of the compression model, demonstrating that FAVICOMP draws on parametric knowledge only when necessary—potentially when the compression model is uncertain about the next token.

# 6 RELATED WORKS

**Evidence Compression for RAG.** Standard RAG retrieves textual evidence related to the prompt from the external corpora or knowledge bases and incorporates it as a part of the input to the LM (Lewis et al., 2020; Izacard & Grave, 2021; Guu et al., 2020). However, retrieved evidence pieces may contain inconsistent or irrelevant information to the question, potentially confusing the target model in downstream tasks (Shi et al., 2023). To tackle this problem, traditional approaches aim to rerank the textual evidence based on its relevance to the question and then select a top-ranked subset to include as part of the input to the LM (Nogueira et al., 2020; Zhuang et al., 2023). However, this approach loses more question-relevant information by discarding lower-ranked sentences.

Recent efforts on evidence compression seek to compress retrieved evidence pieces to filter out unnecessary information and retain only the essential context (Wang et al., 2023c; Li et al., 2024d; Ke et al., 2024; Jiang et al., 2023a; Xu et al., 2024; Cao et al., 2024; Yoon et al., 2024). Wang et al. (2023c) filter query-relevant context using relevance metrics and Li et al. (2024d) extract query-relevant information and restructure them to form a consistent context. Ke et al. (2024) trains a seq2seq bridge model using supervised and reinforcement learning to optimize the connection between the retriever and the LLM. Jiang et al. (2023a) and Cao et al. (2024) conduct token-level

or embedding-based compression to preserve only the query-relevant information using a trained compressor. Xu et al. (2024) and Yoon et al. (2024) train a compression model to generate an abstractive summary of the documents by distilling knowledge from larger language models. While these methods are successful to some extent, they often achieve suboptimal performance because the compressed context may be unfamiliar to the LM used in the downstream task due to differences in pretrained internal knowledge and prompt preferences between the compression and the target model. In contrast, FAVICOMP proactively compresses the evidence pieces in a way to lower the target model's perplexity using an ensemble decoding technique without any training, thereby improving the downstream performance.

**Parametric and Non-parametric Knowledge in RAG.** While there have been studies on the phenomena of LM's utilization of both parametric and non-parametric knowledge sources (Longpre et al., 2021; Wadhwa et al., 2024; Wu et al., 2024; Zhang et al., 2024; Zhou et al., 2023; Wang et al., 2023a; Fang et al., 2024), there is a lack of research focused on effectively synergizing both sources. A few of these efforts introduce counterfactual augmentation (Longpre et al., 2021; Fang et al., 2024; Zhang et al., 2024) and causal intervention (Zhou et al., 2023; Wang et al., 2023a) to mitigate knowledge conflict, which, however, requires explicitly knowing the features of the input that causes such conflict. Zhang et al. (2023) seek to address this issue by incorporating LM-generated context into the LM's input along with the retrieved documents, thereby integrating both sources of knowledge. However, merely concatenating both contexts is a suboptimal solution, as LMs may still show bias toward one source over the other when generating responses (Longpre et al., 2021; Wu et al., 2024). To address this, FAVICOMP employs ensemble decoding during the evidence compression, ensuring that both types of knowledge are seamlessly fused together to create a consistent context.

**Constrained Decoding.** Constrained decoding has been previously proposed in text generation tasks for various purposes, including optimizing prompts (Liu et al., 2024), enhancing plausibility (Li et al., 2023) or controllability (Meng et al., 2022; Huang et al., 2023), and reducing hallucination (Shi et al., 2024). Contrastive Decoding (Li et al., 2023) enforces a plausibility constraint during generation by inducing the difference in token log-probabilities between expert and amateur LMs. Context-aware Decoding (Shi et al., 2024) uses contrastive decoding to amplify the probability differences between outputs with and without evidence, encouraging the LM to prioritize the evidential knowledge. Our work is closely connected with the method by Liu et al. (2024) which employs ensemble decoding to paraphrase prompts to enhance zero-shot LM prompting and generalization. Their approach focuses on the robustness and generalizability of instruction prompts for tasks without retrieval augmentation. In contrast, our approach compresses externally retrieved evidence while integrating parametric knowledge during compression, specifically targeting knowledge-intensive tasks that require balancing both evidential and parametric knowledge.

## 7 CONCLUSION

In this study, we introduce FAVICOMP, a training-free evidence compression method designed to enhance RAG by making retrieved evidence set more familiar to the target model, while seamlessly integrating parametric knowledge. By leveraging ensemble decoding, FAVICOMP compresses the retrieved evidence to make it more favorable to the target model. Moreover, FAVICOMP effectively balances the target model's parametric knowledge and the retrieved knowledge, improving performance on complex tasks where the retrieved evidence set may not contain all the necessary information. Our extensive experiments validate the effectiveness of FAVICOMP on open-domain QA tasks, showing significant improvements over recent evidence compression baselines in multiple datasets. Additionally, FAVICOMP's model-agnostic nature allows it to be effortlessly incorporated into various RAG workflows without additional training, making it a versatile tool for enhancing LMs in complex tasks.

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

## A  IMPLEMENTATION DETAILS

(1) **Gold Compression**: We implement the Gold Compression baseline following the approach outlined by Yoon et al. (2024). We evaluate only on HotpotQA, 2WikiMQA, and MuSiQue, as these datasets contain gold documents. We first identify the presence of any gold documents in the retrieved documents. If found, we use the documents as the context. If none of the retrieved documents are identified as gold, we utilize the entire set of retrieved documents as the context for the evaluation. To identify the gold documents within the retrieved documents, we compare each gold document with the retrieved ones. If 50% or more of the content matches, we classify it as a gold document. This approach is necessary because the documents are chunked, and the retrieved documents may not exactly match the gold documents.

(2) **Generated Context**: We use the context generation prompt in Tab. 6 to generate the context.

(3) **Zero-shot Summarization**: We use the evidence compression prompt in Tab. 6 to compress the retrieved documents.

(4) **RECOMP-extractive**: We utilize the same Contriever models trained by the authors for each dataset, to encode both the question and the sentences in the evidence set. For 2WikiMQA and MuSiQue, since there are no fine-tuned models available, we use the Contriever fine-tuned on HotpotQA. Following the original paper, we select one sentence as the context for NQ and TQA, whereas for the other datasets, we utilize two sentences.

(5) **RECOMP-abstractive**: Similar to RECOMP-extractive, we use the same T5-large models trained by the authors for each dataset to compress the retrieved evidence. For the 2WikiMQA and MuSiQue, we employ the T5-large model fine-tuned on HotpotQA.

(6) **LongLLMLingua**: We use `Llama2-7B`[5] trained by the authors as the prompt compressor model. We use the default hyperparameters in the original paper, where the dynamic context compression rate is set to 0.3, and the maximum compression rate is set to 0.5.

(7) **CompAct**: We use the same `Mistral-7B-Instruct`[6] model instruction-tuned by the authors for evidence compression. The number of documents per segment is set to 5 with 1 iteration.

## B  ADDITIONAL EXPERIMENT RESULTS

### B.1  MISTRAL-7B-INSTRUCT AS COMPRESSION AND TARGET MODEL

We conduct an experiment where we use `Mistral-7B-Instruct` as the compression and target model. The result in Tab. 3 demonstrates that FAVICOMP outperforms all other baselines, supplementing the effectiveness shown in §4.1

### B.2  HEAD-TO-HEAD COMPARISON WITH RECOMP-ABSTRACTIVE

We conduct a head-to-head experiment on RECOMP-abstractive by using the same base compression model as FAVICOMP for a more fair comparison. We construct training data on NQ, TQA, and HotpptQA according to Xu et al. (2024) and finetune `Mistral-7B-Instruct` on each of the training data. We train for 7 epochs using LoRA with Adam optimizer with a learning rate of

---

[5]https://huggingface.co/NousResearch/Llama-2-7b-hf
[6]https://huggingface.co/cwyoon99/CompAct-7b

| Methods | Size | NQ | | TQA | | HotpotQA | | 2WikiMQA | | MuSiQue | |
|---------|------|------|------|------|------|------|------|------|------|------|------|
| | | Acc | F1 | Acc | F1 | Acc | F1 | Acc | F1 | Acc | F1 |
| *Mistral-7B-Instruct* | | | | | | | | | | | |
| Gold Document | | - | - | - | - | 41.0 | 50.5 | 38.1 | 40.3 | 9.6 | 15.2 |
| No Context | | 28.1 | 27.5 | 58.8 | 60.9 | 19.7 | 24.8 | 21.9 | 22.8 | 5.2 | 9.7 |
| Raw Document | | 40.2 | 39.3 | **66.2** | 68.6 | 30.3 | 37.2 | 26.6 | 28.5 | 7.5 | 13.1 |
| Generated Context | | 30.1 | 31.7 | 57.3 | 60.7 | 23.7 | 30.6 | 25.1 | 29.5 | 7.1 | 12.8 |
| Sentence-BERT | 110M | 29.8 | 30.1 | 57.8 | 60.7 | 23.8 | 30.3 | 22.9 | 24.7 | 7.5 | 12.3 |
| RECOMP-extractive | 110M[†] | 31.7 | 32.2 | 57.2 | 60.0 | 24.1 | 30.2 | 23.2 | 24.4 | 7.4 | 12.5 |
| LongLLMLingua | 7B[†] | 34.3 | 36.4 | 63.8 | 66.9 | 27.0 | 34.7 | 25.5 | 28.0 | 7.1 | 13.0 |
| RECOMP-abstractive | 775M[†] | 38.0 | 37.8 | 62.1 | 65.0 | 27.4 | 34.3 | 25.1 | 27.4 | 6.4 | 12.0 |
| CompAct | 7B[†] | 38.8 | 38.9 | 65.1 | 67.1 | 30.2 | 37.1 | 24.9 | 27.6 | 8.2 | 13.6 |
| Zero-shot Summarization | 7B | 38.4 | 38.2 | 62.3 | 64.8 | 28.2 | 35.2 | 23.2 | 27.1 | 6.8 | 11.8 |
| **FAVICOMP** | 7B | **40.3** | **40.4** | 65.9 | **68.9** | **32.0** | **40.5** | **29.7** | **35.1** | **9.2** | **15.2** |

Table 3: Experimental results when FAVICOMP has different compression and target models. We test using `Mixtral-8x7B-Instruct` as the target model on five open-domain QA datasets across all the methods. `Mistral-7B-Instruct` is used as the compression model of FAVICOMP. The best Accuracy and token-level F1 scores for each dataset are in bold.

2e-6 and a batch size of 64. We present the evaluation results in Tab. 4. Even though using larger base model for compression enhances the performance of RECOMP-abstractive to some extent, it still underperforms compared to training-free FAVICOMP. This underscores that the familiarization during evidence compression and integration of parametric and non-parametric knowledge are more helpful to the downstream generation than relying on a trained model for evidence compression.

| Methods | Train | Compression Model | NQ | | TQA | | HotpotQA | |
|---------|-------|-------------------|------|------|------|------|------|------|
| | | | Acc | F1 | Acc | F1 | Acc | F1 |
| RECOMP-abstractive | O | T5-large | 38.0 | 37.8 | 62.1 | 65.0 | 27.4 | 34.3 |
| RECOMP-abstractive | O | Mistral-7B-Instruct-v0.3 | 38.3 | 38.2 | 63.0 | 65.4 | 29.5 | 36.6 |
| FaviComp | X | Mistral-7B-Instruct-v0.3 | 40.3 | 40.4 | 65.9 | 68.9 | 32.0 | 40.5 |

Table 4: Head-to-head comparison results with RECOMP

## C  PROMPT TEMPLATES

### C.1  EVALUATION

The evaluation prompt template is shown in Fig. 5. For all the evaluations throughout the experiment, we switch the positions of the Question and Context if doing so results in better performance. System prompts and demonstrations used in the evaluation are presented in Tab. 5 and Tab. 7, respectively.

### C.2  FAVICOMP

The prompt templates for evidence compression and context generation of FAVICOMP are presented in Tab. 6

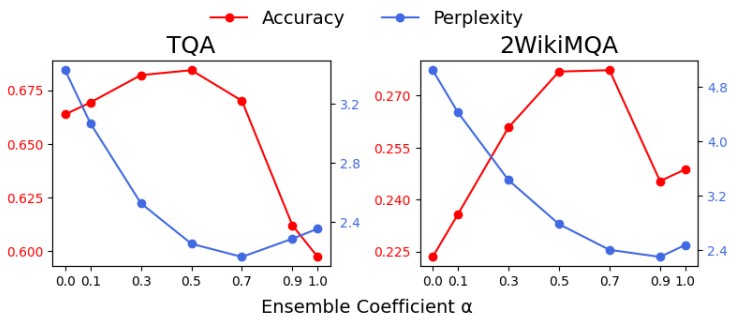

Figure 4: Impact of coefficient $\alpha$ on performance and perplexity for TQA and 2WikiMQA.

---

**Evaluation Prompt Template**

```
{System Prompt}
{Demonstrations}
Question: {Question}
Context: {Context}
Answer:
```

---

Figure 5: Evaluation Prompt Template.

| Model | System Prompt |
|---|---|
| Llama-3-8B-Instruct | You are an expert in Question Answering. Your job is to answer questions in 1 to 5 words based on the given context. |
| Mistral-7B-Instruct | You are an expert in Question Answering. Your job is to answer questions in 1 to 5 words based on the given context. Just output the answer as concisely as possible, no other words |

Table 5: System prompts used in evaluation

| Instruction | Prompt Template |
|---|---|
| Evidence Compression | You are an expert in summarization. Given a question and multiple document snippets, generate one summarized context that is helpful to answer the question. Just summarize, no other words.
Question: {Question}
Documents: {Evidence}
Summarized Context: |
| Context Generation | You are an expert in context generation. Given a question, generate a context that is helpful to answer the question. Just generate the context, no other words.
Question: {Question}
Context: |

Table 6: Prompt Templates for FAVICOMP

| Dataset | Demonstrations |
|---------|----------------|
| NQ | Question: who sings i've got to be me
Answer: Sammy Davis, Jr
Question: who wrote i will follow you into the dark
Answer: Ben Gibbard
Question: who won season 2 of total drama island
Answer: Owen (Scott McCord)
Question: what part of the mammary gland produces milk
Answer: cuboidal cells
Question: when did the golden compass book come out
Answer: 1995 |
| TQA | Question: Who sang the theme for the James Bond film 'Thunderball'?
Answer: Tom Jones
Question: A hendecagon has how many sides?
Answer: Eleven
Question: In the 1968 feature film Chitty Chitty Bang Bang, of what country is Baron Bomburst the tyrant ruler?
Answer: Vulgaria
Question: Artists Chuck Close, Henri-Edmond Cross, John Roy, Georges-Pierre Seurat, Paul Signac, Maximilien Luce and Vincent van Gogh painted in what style?
Answer: Pointillism
Question: What is the study of the relation between the motion of a body and the forces acting on it?
Answer: Dynamics |
| HotpotQA | Question: Which magazine was started first Arthur's Magazine or First for Women?
Answer: Arthur's Magazine
Question: The Oberoi family is part of a hotel company that has a head office in what city?
Answer: Delhi
Question: Musician and satirist Allie Goertz wrote a song about the "The Simpsons" character Milhouse, who Matt Groening named after who?
Answer: President Richard Nixon
Question: Are Jane and First for Women both women's magazines?
Answer: Yes
Question: Were Pavel Urysohn and Leonid Levin known for the same type of work?
Answer: No |
| 2WikiMQA | Question: Where was the place of death of Marie Thérèse Of France (1667–1672)'s father?
Answer: Palace of Versailles
Question: Who is the paternal grandmother of Przemysław Potocki?
Answer: Ludwika Lubomirska
Question: Who lived longer, Herbert Findeisen or Léonie Humbert-Vignot?
Answer: Léonie Humbert-Vignot
Question: Are Alison Skipper and Diane Gilliam Fisher from the same country?
Answer: Yes
Question: Are director of film Move (1970 Film) and director of film Méditerranée (1963 Film) from the same country?
Answer: No |
| MuSiQue | Question: Who is the child of the director and star of Awwal Number?
Answer: Suneil Anand
Question: What is the record label of the rapper who performed Jigga My?
Answer: Roc-A-Fella Records
Question: What county shares a border with the county where Black Hawk Township is located?
Answer: Dodge County
Question: Who is the sibling of the person credited with the reinvention and popularization of oil paints?
Answer: Hubert Van Eyck
Question: Who heads the Catholic Church, in the country that a harp is associated with, as a lion is associated with the country that Queen Margaret and her son traveled to?
Answer: Eamon Martin |

Table 7: Demonstrations used in evaluation for each dataset

