# OpenReview forum: "Familiarity-Aware Evidence Compression for Retrieval-Augmented Generation"
_ICLR.cc/2025/Conference — ICLR 2025 Conference Withdrawn Submission_

### Official Review · Reviewer_H18R · 2024-10-16

**Soundness:** 2
**Presentation:** 3
**Contribution:** 2
**Rating:** 5
**Confidence:** 4

**Summary:**

The authors propose a training-free evidence compression technique named FaviComp. Specifically, FaviComp actively combines compressed evidence to reduce the perplexity of the target model by combining the decoding probabilities of the compression model and the target model to generate a more familiar context for the target model. Experimental results show that FaviComp consistently outperforms the state-of-the-art evidence compression baselines on multiple open-domain QA datasets.

**Strengths:**

1. An effective prompt compression method is proposed, which achieves the optimal compression ratio on multiple datasets while improving the performance indicators of the model.
2. No training is required, and compression is performed during inference, which reduces the training cost, but there is also a concern that the inference time will increase.
3. The paper is generally clear and the presentation is generally clear.

**Weaknesses:**

1. Some details need to be clarified. The theoretical basis and intuition of the motivation need to be further elaborated.
2. The novelty of the method is limited. In essence, it is more like an ensemble that generates different evidences at different prompts, using the target LLM to generate evidence and the original document to generate evidence.
3. Experimental ablation needs to be increased.

**Questions:**

1. What if the target LLM is considered during compression and the target LLM is in the form of an API or uncertain?
2. The author should rigorously define what is evidence that the target LLM is familiar with. In addition, if only the target LLM is familiar with it, does familiarity necessarily mean high compression quality? Does familiarity necessarily mean that the target LLM will use the compressed evidence to the greatest extent? What is the theoretical basis for the starting point?
3. Ablation: (1) Only allowing the target LLM to provide evidence (that is, only the formula of L181, that is, 2-time inference) will exceed the raw document in 2wikiMQA, which is inconsistent with other datasets. (2) What if the compression model is the target LLM? This situation implies that the compression evidence is familiar with the target LLM. In fact, referring to L231-L236, the target LLM and the compressed LLM are consistent (so I question the starting point of L49-L52. In essence, it is more like an ensemble in which the prompt generates different evidences at different times, using the target LLM to generate evidence and the original document to generate evidence respectively). (3) In the Llama model, have you ever tried the situation where the compressed LLM and the target LLM are inconsistent? The parameter size is inconsistent or the model source is inconsistent. (4) L307-L309, why can the performance of the model exceed the performance on the gold document after compression? Shouldn't this be the upper limit? What is the reason for this phenomenon? (5) How does the experiment of concating the compressed texts generated separately from the two parts perform?
5. Considering that the method uses the internal knowledge of LLM when regenerating evidence (which may increase the number of reasoning steps), I hope to see some analysis of the corresponding multi-document and multi-hop QA advantages.

---

> ### Author Response · Authors · 2024-11-23
> **Response to Reviewer H18R (1/2)**
>
> We sincerely thank the reviewer for their thoughtful feedback and constructive suggestions. Below, we address your concerns and clarify key points.
>
> ### **1. Regarding the target LLM is the form of API or uncertain**:
> We consider black-box settings where we only have access to output logits from the models. If the API provides the output logits of the prediction, we can use FaviComp. Also, since we ensemble the token logits from the compression and the target model, if the target model exhibits uncertainty on the next token probability, the token probabilities from the compression model supplement the limited knowledge of the target model. Thus, we could integrate both sources of knowledge in synergy.
>
> ### **2. Regarding the definition of familiarity of an evidence to the target LLM**:
> We want to clarify that we have defined the familiarity of evidence in terms of perplexity in L76 and L166. In brief, the lower the compressed evidence’s perplexity, the more familiar to the target model. It has been proven from previous works that the lower perplexity of a prompt means that the model is better able to generalize to the downstream task [1,2,3]. We build upon these findings. We compress the raw evidence into a compact form that exhibits lower perplexity (or familiarity) to the target model to enhance the performance of the target model.
>
> [1] Liu et al. Monotonic Paraphrasing Improves Generalization of Language Model Prompting. EMNLP 2024.
>
> [2] Gonen et al. Demystifying Prompts in Language Models via Perplexity Estimation. EMNLP 2023 findings.
>
> [3] Wang et al. ReadPrompt: A Readable Prompting Method for Reliable Knowledge Probing. EMNLP 2023 findings.
>
> ### **3. Regarding the inconsistency in the experiment:**
> Since every target model has different internal knowledge and prompt preference, it is hard to tell what is the reason for the lower performance of Raw Document than Generated Context for Llama3-8B-Instruct. However, we believe that it is because 1) Llama3-8B-Instruct contains more knowledge to answer 2WikiMQA with its parametric knowledge than the other target model (Mistral-7B) and 2) since 2WikiMQA is a multi-hop QA dataset and the retrieval quality is low, Raw Document baseline did not provide more relevant information in the context than other datasets.
>
> ### **4. Regarding the use of the same model as compression and target mode**:
> We acknowledge that using the same models as compression and target models makes the compressed evidence more familiar than using different models. However, even though the compression and target models are the same, depending on the compression methods, one method might exhibit more familiarity towards the target model. As shown in Table 3 of case study, FaviComp exhibits lower perplexity than Zero-shot Summarization, even though they both use the same compression and target models. This is because the context generated from the target model in FaviComp makes the compressed evidence more favorable to the target model. On the other hand, Zero-shot Summarization only focuses on choosing tokens that summarize the external documents without considering familiarity.
>
> To make our point more clear, we additionally conducted experiments with Llama3-8B-Instruct using smaller Llama3.2-3B-Instruct as compression model that is different from the target model. The result is in our official comment above here (https://openreview.net/forum?id=Zd8ODMYMBZ&noteId=moSFzkRHFo). We have updated the full evaluation results in Table 1 of our revised draft.
>
> Despite using different and smaller compression model, it performs comparably with using the same Llama3-8B-Instruct as compression and target model, which proves that FaviComp algorithm is the main reason for making evidence compression familiar to the target model.

---

> ### Author Response · Authors · 2024-11-23
> **Response to Reviewer H18R (2/2)**
>
> ### **5. Regarding the use of different compression and target model for Llama**:
> We have additionally experimented with Llama3-8B-Instruct using a smaller Llama3.2-3B-Instruct as the compression model as mentioned in the official comment (https://openreview.net/forum?id=Zd8ODMYMBZ&noteId=moSFzkRHFo). We have updated the full results in Table 1 of our revised draft.
>
> ### **6. Regarding exceeding the performance of Gold Document**
> We want to note that the Gold Document setting considers the gold documents within the retrieved documents. Since we retrieved top-5 documents from the Wikipedia corpus, the retrieved documents might contain not all or even none of the gold documents. If none of the retrieved documents are identified as gold, we utilize the entire set of retrieved documents as the context for the evaluation. So the setting is considered as perfect compression within top-k retrieved document. We chose this setting following the previous work [1]. We have illustrated this setting in L713 and changed the name to “Gold Compression” to avoid confusion in our revised draft.
>
> ### **7. Regarding the performance of concatenating compressed and generated context**:
> We experimented with concatenating compressed text (using Zero-shot Summarization baseline) with generated context (using Generated Context baseline). The result is as follows.
>
> |                         |             |             |             |             |            |
> | ----------------------- | ----------- | ----------- | ----------- | ----------- | ---------- |
> | Method                  | NQ          | TQA         | HotpotQA    | 2WikiMQA    | MuSiQue    |
> | Generated Context       | 32.3 / 36.6 | 59.7 / 62.4 | 22.7 / 29.7 | 24.8 / 28.7 | 7.6 / 14.8 |
> | Zero-shot Summarization | 41.3 / 45.1 | 66.3 / 69.5 | 30.2 / 38.6 | 22.3 / 28.1 | 8.3 / 16.3 |
> | Concat                  | 38.5 / 42.1 | 66.5 / 68.9 | 27.8 / 35.9 | 20.6 / 25.7 | 8.5 / 16.7 |
>
> We have observed that it consistently underperforms compared to the Zero-shot Summarization baseline except for MuSiQue dataset. This implies that simply concatenating non-parametric knowledge with parametric knowledge is actually worse than just using non-parametric knowledge in the context. On the other hand, FaviComp seamlessly fuses both knowledge sources during compression to utilize both knowledge in synergy.
>
> ### **8. Regarding the advantage of multi-document or multi-hop QA**:
> Our method is advantageous in multi-document and multi-hop QA settings (or any complex QA tasks that may benefit from multi-evidence) where the retrieval quality is low since we seamlessly inject parametric knowledge during evidence compression. In section 4.3, for multi-document and multi-hop QA datasets (HotpotQA, 2WikiMQA, MuSiQue), we analyzed how the integration of internal knowledge helps answering the question with inaccurate retrieval (HITS=0) and also with accurate retrieval (HITS=1). We believe the existing analyses have provided a comprehensive illustration of FaviComp’s merits from a few distinct aspects. Please let us know if there are any other specific analyses that you want to see regarding multi-document and multi-hop QA settings.
>
> We hope our responses have addressed your concerns and clarified the contributions of our work. We are deeply grateful for your thoughtful feedback and kindly request you to consider raising your score if our clarifications meet your expectations. Thank you again for your time and valuable insights.
>
> [1] Yoon et al. CompAct: Compressing Retrieved Documents Actively for Question Answering

---

> ### Comment · Reviewer_H18R · 2024-11-25
>
> Thank you for the detailed response, which addressed most of my questions. However, I am still concerned about the theoretical basis and intuition of the motivation of the proposed method. And I read the comments of other reviewers and also recognized the limitations of inference time and vocabulary. Therefore, I maintain my score unchanged.

---

> ### Author Response · Authors · 2024-11-29
> **Thank you for your feedback reviewer H18R!**
>
> Dear reviewer H18R,
>
> Thank you for your feedback! We appreciate your constructive reviews for improving our work.
>
> Regarding your concern on "the theoretical basis and intuition of the motivation", as we mentioned in 2. of our response, our work's theoretical basis is based on the previous findings that **"LM’s familiarity with a prompt is generally reflected by low perplexity and the lower perplexity of a prompt means that the model is better able to generalize to the downstream task"** [1,2,3]. Thus, our primary goal is to **lower the perplexity of the multiple evidence during compression to enhance performance**. Also, we have illustrated the motivation in section 2.1 and Figure 1 of the paper. **Please let us know if there are any other clarifications needed to address the lack of the theoretical basis and intuition of the motivation.**
>
> Also, we believe that our method has more advantages than limitations regarding **enhancing the performance of knowledge-intensive tasks that require consolidating multiple pieces of evidence**. We will try to mitigate the limitations in our future revisions. Please let us know if you have any additional questions or concerns we might want to address.
>
>
> [1] Liu et al. Monotonic Paraphrasing Improves Generalization of Language Model Prompting. EMNLP 2024.
>
> [2] Gonen et al. Demystifying Prompts in Language Models via Perplexity Estimation. EMNLP 2023 findings.
>
> [3] Wang et al. ReadPrompt: A Readable Prompting Method for Reliable Knowledge Probing. EMNLP 2023 findings.

---

### Official Review · Reviewer_XDL4 · 2024-10-16

**Soundness:** 2
**Presentation:** 2
**Contribution:** 1
**Rating:** 3
**Confidence:** 4

**Summary:**

This paper proposes FAVICOMP, a training-free evidence compression technique that makes retrieved evidence more familiar to the target model, while seamlessly integrating parametric knowledge from the model.
FAVICOMP leverages the decoding probabilities of two LMs, a compression model and the target model. The compression model is instructed to summarize the raw evidential documents into a relevant context to the input, while the target model is instructed to generate relevant context without referencing the documents.
By combining decoding probabilities from both the compression model and the target model, FAVICOMP can generate context that is more familiar to the target model.
Experiments on open-domain QA demonstrate that FAVICOMP shows better QA performance.

**Strengths:**

1. Compression-based RAG presents a promising research direction in terms of reducing inference latency and improving robustness to irrelevant retrieved evidence.

2. This paper proposes an interesting perspective of enabling the target model to become familiar with the compressed summary.

**Weaknesses:**

1. The motivation of compression-based RAG is to improve the latency of standard RAG. However, ensembling both the decoding probabilities of the compression model and the target model does not save the inference costs compared to standard RAG. I'm not sure whether FAVICOMP even consumes more computation than standard RAG.

2. I have doubts about the validity of the assumption that ensembling decoding probabilities under different conditions. The compression model is decoding under line 159: (Evidence Compression Instruction, input question, retrieved documents), while the target model is decoding under line 180: (Context Generation Instruction, input question). Although authors explain that both objectives ultimately share the goal of generating context relevant to the question, this is too inconsistent in formulation.

3. Since compression-based RAG is to improve the latency of standard RAG, the compression model should be more lightweight than the target model. For example, RECOMP adopts T5 (770M) as the compression model and Flan-UL2 as the target model (20B). However, the compression models are the same as the target models for FAVICOMP, which does not make sense.

4. Since FAVICOMP does not use a compression model of the same scale, the results in Table 1 are unfair. FAVICOMP would likely achieve better results, as it employs a more powerful compression model.

5. Reporting only the QA results in Table 1 is insufficient, as the compression rate is just as important as accuracy and F1. There’s a trade-off between compression rate and QA performance, so it would be better to combine Table 1 and Table 2. It’s difficult to assess the effectiveness of FAVICOMP without considering the compression rate.

**Questions:**

See above.

---

> ### Author Response · Authors · 2024-11-23
> **Response to Reviewer XDL4 (1/2)**
>
> We sincerely thank the reviewer for their thoughtful feedback and constructive suggestions. Below, we address your concerns and clarify key points.
>
> ###  **1. Regarding the latency of FaviComp**:
> We want to clarify that our contribution is to address the limitation of standard RAG and compression-based RAG, where the compressed evidence is suboptimal to the target model due to its unfamiliarity. We do acknowledge that it consumes more computation during compression since we infer two models to predict the next token probabilities, but we want to emphasize that our method is a training-free method that can familiarize evidence and integrate parametric knowledge without training compression or target model.
>
> ###  **2. Regarding the validity of ensembling decoding probabilities**:
> Previous works on constrained decoding have successfully proved the validity of ensembling or contrasting decoding probabilities under different conditions [1,2,3]. Also, since our instructions are very similar in that they explicitly state to output context that is helpful to answer the question, we did not encounter any conflicts between these two decodings. We apologize for the confusion from insufficient information on the validity of the ensemble decoding.
>
> [1] Kim et al. Instructive Decoding: Instruction-Tuned Large Language Models are Self-Refiner from Noisy Instructions. ICLR 2024.
>
> [2] Shi et al. Decoding-Time Language Model Alignment with Multiple Objectives. ICML 2024 Workshop.
>
> [3] Shi et al. Trusting Your Evidence: Hallucinate Less with Context-aware Decoding. NAACL 2024 short.
>
> ###  **3. Regarding the use of the same model of compression and target models**:
> We have addressed the cases where the compression model is much smaller than the target model, where it uses Mistral-7B-Instruct as the compression model and Mixtral-8x7B-Instruct as the target model. Since the result table was in the appendix, we moved the results to Table 1 in our updated draft for better visibility.
>
> Also, we conducted an additional experiment using Llama as mentioned in the above official comment (https://openreview.net/forum?id=Zd8ODMYMBZ&noteId=moSFzkRHFo), where we used Llama-3.2-3B-Instruct as the compression model and Llama3-8B-Instruct as the target model. The full results is in Table 1 in our revised draft.

---

> ### Author Response · Authors · 2024-11-23
> **Response to Reviewer XDL4 (2/2)**
>
> ### **4. Regarding the use of smaller compression models for the baselines:**
>
> Even though it is not a head-to-head comparison with baselines, we wanted to compare how our **training-free** method performs compared to the methods that use **fully-supervised** compression models that have shown promising performance. For instance, RECOMP [1] demonstrated their trained T5-large compression model performing comparably with the GPT-3.5 compressor in Table 2 of their paper.
>
> To address your concerns about the fairness of the experiment, we conducted an additional experiment with RECOMP-abstractive by replacing the original T5-large with newly finetuned Mistral-7B-Instruct-v0.3 as the compression model for each dataset: NQ, TQA, and HotpotQA. We use the base Mistral-7B-Instruct-v0.3 for answer generation. The results are in the above official comment (https://openreview.net/forum?id=Zd8ODMYMBZ&noteId=moSFzkRHFo). We present the full evaluation settings and results in Appendix B.2 and Table 4 of the revised draft.
>
> ### **5. Regarding the combining evaluation results with compression rates:**
>
> Thank you for the advice. We do think that it will be better to put the QA performance and the compression rate together for easier comparison. We have incorporated compression rates in Table 1 in our revised draft.
>
> We hope our responses have addressed your concerns and clarified the contributions of our work. We are deeply grateful for your thoughtful feedback and kindly request you to consider raising your score if our clarifications meet your expectations. Thank you again for your time and valuable insights.
>
> [1] Xu et al. Recomp: Improving retrieval-augmented lms with context compression and selective augmentation. ICLR 2024.

---

> > ### Comment · Reviewer_XDL4 · 2024-11-25
> >
> > Thanks for the response.
> >
> > My concern regarding the validity of ensembling decoding probabilities has been addressed.
> >
> > However, my concerns regarding the latency and effectiveness still exist. The authors' response to reviewer CjL3 confirms that the inference cost of the proposed solution is twice that of a traditional RAG solution. Although the proposed method is training-free and can achieve better performance, it contradicts with the motivation that **compression is to improve latency and reduce cost first, not mandatory to improve QA performance as well (but it would be much better if more effective and efficient)**. I do not think it is a good idea to put this work under the story of compression-based RAG. It would make more sense to just say **we propose a training-free method that can enhance the performance of standard RAG with double costs**.
> >
> > Therefore, I will maintain my score.

---

> > > ### Author Response · Authors · 2024-11-29
> > > **Thank you for your feedback reviewer XDL4!**
> > >
> > > Dear reviewer XDL4,
> > >
> > > Thank you for your feedback! We appreciate your constructive reviews for improving our work.
> > >
> > > We wanted to clarify that, unlike previous RAG methods that use prompt compression, our primary goal of using compression was to **consolidate multiple pieces of evidence of knowledge-intensive tasks.** Even though our method is actually compressing multiple pieces of evidence, the purpose of the compression is to **consolidate multiple pieces of evidence.**
> > >
> > > Also, thank you for your suggestion to improve our story. We will enhance our story to focus on consolidating multiple pieces of evidence rather than "compressing" them. We believe this shift in focus will better highlight our goal of improving the performance of RAG instead of merely reducing its cost. Please let us know if you have any additional questions or concerns we might want to address.

---

### Official Review · Reviewer_CjL3 · 2024-11-02

**Soundness:** 3
**Presentation:** 3
**Contribution:** 3
**Rating:** 6
**Confidence:** 3

**Summary:**

This paper introduces FAVICOMP (Familiarity-aware Evidence Compression), a training-free model designed to make retrieved evidence more familiar to a target model. FAVICOMP compresses evidence by minimizing perplexity, combining decoding probabilities from both the compression model and the target model to generate context that aligns more closely with the target model’s familiarity. Experimental results indicate that FAVICOMP outperforms baseline approaches.

**Strengths:**

This paper presents a training-free solution for adapting retrieved evidence to a target model, an interesting and innovative direction.

Experimental results on LLaMA 3 8B Instruct and Mistral 7B Instruct demonstrate that the proposed solution outperforms trained ranker and compressor models.

**Weaknesses:**

While empirical results are provided, the intuitive design that relies on the perplexity of both the compressor and target models requires further justification. Given that target models often have limitations such as restricted knowledge and tendencies toward hallucination, it would be beneficial to address how these limitations impact the proposed approach’s effectiveness.

Missing discussion of related work:
Bridging the Preference Gap between Retrievers and LLMs

**Questions:**

What is the inference efficiency (in terms of computational cost) of the proposed solution?

---

> ### Author Response · Authors · 2024-11-23
> **Response to Reviewer CjL3**
>
> We sincerely thank the reviewer for their thoughtful feedback and constructive suggestions. Below, we address your concerns and clarify key points.
>
> ### **1. Regarding justification of relying on the perplexity of both the compressor and target models:**
> We want to clarify that our method alleviates the limitation of standard RAG of over or under-relying on the given context by balancing the retrieved knowledge and the parametric knowledge during evidence compression. If the retrieval accuracy is low, the token probabilities from the target model’s context generation supplement the token probabilities from inaccurate evidence compression. On the other hand, if the target model exhibits uncertainty on the next token probability, the token probabilities from the compression model supplement the limited knowledge of the target model. This is visualized in Table 2 of our paper.
>
> ### **2. Regarding the missing discussion of related work.**
> Thanks for your suggestion of the paper “Bridging the Preference Gap between Retrievers and LLMs”. We have included the work in our related section in the revised draft in L484-485.
>
> ### **3. Regarding the inference efficiency:**
> For each decoding step, we infer two models to predict the next token probabilities. Thus, the inference cost would be roughly twice as much compared to using one compression model. Since we use cache for previous inputs and generations for both compression and target model, there are no additional costs for prefilling the cache for each decoding step. However, if the inference cost is not the utmost priority, we believe that our method has much more benefits, 1) optimizing the evidence to be more preferable to the target model and 2) introducing parametric knowledge to mitigate imprecise retrieval.
> In addition, we could potentially reduce inference cost by using parallel computation for the compression and the target model.
>
> We hope our responses have addressed your concerns and clarified the contributions of our work. We are deeply grateful for your thoughtful feedback and kindly request you to consider raising your score if our clarifications meet your expectations. Thank you again for your time and valuable insights.

---

> > ### Comment · Reviewer_CjL3 · 2024-11-24
> > **Acknowledging the Authors' Response**
> >
> > Thank you to the authors for their detailed response. The response confirms that the inference cost of the proposed solution is twice that of a traditional RAG solution. However, the proposed solution is training-free, which could reduce training costs and provide flexibility in combining compression models with target models.
> >
> > I would like to maintain my original score.

---

> > > ### Author Response · Authors · 2024-11-29
> > > **Thank you for your feedback reviewer CjL3!**
> > >
> > > Dear reviewer CjL3,
> > >
> > > Thank you for your feedback! We appreciate your constructive reviews for improving our work.
> > >
> > > Although the inference cost is larger than the baselines, we believe that our method has more advantages than limitations regarding **enhancing the performance of knowledge-intensive tasks that require consolidating multiple pieces of evidence**. We will try to mitigate the limitations in our future revisions. Please let us know if you have any additional questions or concerns we might want to address.

---

### Official Review · Reviewer_pqra · 2024-11-05

**Soundness:** 3
**Presentation:** 3
**Contribution:** 2
**Rating:** 5
**Confidence:** 4

**Summary:**

This work introduces a training-free ensemble decoding algorithm named FaviComp for content compression in retrieval-augmented generation (RAG). It samples tokens from both the compression model and the target model to generate a summary of retrieved documents, aiming to filter out irrelevant information in retrieved documents and integrate external knowledge with the LLM’s internal parametric knowledge to improve accuracy. Experiments on five QA tasks show that the proposed FaviComp achieves competitive results compared to selected baselines.

**Strengths:**

- The idea of incorporating retrieved external knowledge with parametric knowledge via ensemble decoding is interesting.
- The proposed FaviComp is a training-tree method and can be easily applied to existing LLMs for content compression in retrieval-augmented generation (RAG).
- FaviComp achieves competitive performance compared to baseline methods across five QA benchmarks.

**Weaknesses:**

- Unfair comparison with baseline models. Despite using the same backbone model as the target model, the proposed FaviComp uses more advanced Llama3-8B-instruct/Mistral-7B-Instruct as the compression model, while the baselines use less capable models, such as T5-large or Llama2-7B, for compression. What is the rationale behind this inconsistent choice? Can we use the same backbone model (e.g., Llama3-8B-instruct/Mistral-7B-Instruct) for compression in both the baseline methods (e.g., LongLLMLingua, RECOMP) and FaviComp to facilitate a fair comparison? Otherwise, this discrepancy can lead to a nontrivial impact on the evaluation results, making the experimental findings less convincing—it is unclear whether the improvement is due to an advanced compression model or the method’s design of FaviComp.
- Unfocused problem setup and lack of justification for using compression in RAG. According to Table 1, almost all compression-based models consistently underperform the naive RAG method (namely, “Raw Document”), which simply takes raw documents as input without filtering or compression. Does this imply that compression may actually be unnecessary in your problem setting? Typically, compression is introduced to filter out irrelevant information for improved accuracy or to reduce redundant tokens in the prompt for more efficient inference without significantly compromising accuracy [1,2]. If efficiency is the priority, a problem setting with a larger number of retrieved documents (e.g., k=20) would make more sense than the current setting. However, in this work, only k=5 documents are used. Does this suggest that the role of compression here is primarily focused on denoising the retrieved documents? Unfortunately, there is no comparison with RAG works specifically focused on denoising [3,4,5] in the experiment section. Can you clarify the purpose of compression in this work and justify its effectiveness in a more focused setting (e.g., k=30 for long-context RAG to assess the compression ability of FaviRAG and/or comparing with denoising RAG methods to evaluate the denoising ability of FaviRAG)?

**References**
1. xRAG: Extreme Context Compression for Retrieval-augmented Generation with One Token. NeurIPS 2024.
2. Context Embeddings for Efficient Answer Generation in RAG. arXiv 2024.
3. Certifiably Robust RAG against Retrieval Corruption. arXiv 2024.
4. InstructRAG: Instructing Retrieval-Augmented Generation via Self-Synthesized Rationales. arXiv 2024.
5. Making Retrieval-Augmented Language Models Robust to Irrelevant Context. ICLR 2024.

**Questions:**

Please address the questions in Weaknesses.

Potential limitation: Does the method require the compression model and the target model to be from the same family, sharing an identical tokenizer? Otherwise, how would ensemble decoding be achieved if their vocabularies differ?

---

> ### Author Response · Authors · 2024-11-23
> **Response to Reviewer pqra (1/2)**
>
> We sincerely thank the reviewer for their thoughtful feedback and constructive suggestions. Below, we address your concerns and clarify key points.
>
> ### **1. Regarding your concerns about using smaller compression models for the baselines**:
>
> Even though it is not a head-to-head comparison with baselines, we wanted to compare how our **training-free** method performs compared to the methods that use **fully-supervised** compression models that have shown promising performance. For instance, RECOMP [1] demonstrated their trained T5-large compression model performing comparably with the GPT-3.5 compressor in Table 2 of their paper.
>
> To address your concerns about the fairness of the experiment, we conducted additional experiment with RECOMP-abstractive by replacing original T5-large with newly finetuned Mistral-7B-Instruct-v0.3 as the compression model for each dataset: NQ, TQA, and HotpotQA. We use the base Mistral-7B-Instruct-v0.3 for answer generation. The results are in the above official comment (https://openreview.net/forum?id=Zd8ODMYMBZ&noteId=moSFzkRHFo). We present the full evaluation settings and results in Appendix B.2 and Table 4 of the revised draft.
>
> RECOMP-abstractive with Mistral-7B-Instruct-v0.3 based compressor slightly performed better than the original RECOMP but the performance increment is not dramatic. This proves that the superior performance of FaviComp is mostly due to the familarized context and the integration of the parametric knowledge, rather than the capability of the compressor.
>
> [1] Xu et al. Recomp: Improving retrieval-augmented lms with context compression and selective augmentation. ICLR 2024.

---

> ### Author Response · Authors · 2024-11-23
> **Response to Reviewer pqra (2/2)**
>
> ### **2. Clarification of the purpose of compression**:
> We wanted to clarify that the purpose of evidence compression in our work is not focused on increasing the efficiency of the inference by evidence compression but more on **increasing the accuracy of the downstream generation** by 1) familiarizing evidence compression to the target model 2) integrating target model’s parametric knowledge to mitigate errors from imprecise retrieval. Thus, the number of retrieved evidence was not a primarily important factor, so we set k=5 which is used in previous work such as RECOMP.
>
> ### **3. Regarding the lack of justification for using compression in RAG**:
> The fact that almost all previous compression-based methods (which use different compression and target models e.g. LongLLMLingua, RECOMP, CompAct) underperform and the fact that Zero-shot Summarization (which has the same compression and target model) perform comparably well indicates that there is likely information loss when using compressed evidence from a compression model that differs from the target model. This result implies that existing compression-based methods that only focus on filtering out irrelevant information or reducing the inference prompt cost are likely to be suboptimal due to the unfamiliarity to the target model that comes from the discrepancy between the compression and the target model. This is exactly the problem we wanted to address in this work.
>
> ### **4. Regarding comparison with denoising baselines:**
> Since our work specifically aims to mitigate the unfamiliarity problem of evidence compression, the scope of the work is limited to evidence compression methods. Since we do not claim that the evidence compression methods are better than other works like [2], which trains the downstream generator to make it robust to irrelevant context, we do not include those lines of work as baselines.
>
> ### **5. Sharing the tokenizer:**
> Yes, both the compression model and the target model have to share the same tokenizer to enable ensemble decoding, which is a limitation of our work.
>
> We hope our responses have addressed your concerns and clarified the contributions of our work. We are deeply grateful for your thoughtful feedback and kindly request you to consider raising your score if our clarifications meet your expectations. Thank you again for your time and valuable insights.
>
> [2] Yoran et al. Making Retrieval-Augmented Language Models Robust to Irrelevant Context. ICLR 2024.

---

> > ### Comment · Reviewer_pqra · 2024-11-25
> >
> > I appreciate the authors’ efforts in providing additional clarifications and experimental results. However, the scope of this paper is limited to compression-based RAG methods and imposes additional requirements on the selection of the compression model and target model (they must share the same tokenizer), which limits its impact and practical value. Therefore, I tend to retain my score.

---

> > > ### Author Response · Authors · 2024-11-29
> > > **Thank you for your feedback reviewer pqra!**
> > >
> > > Dear reviewer pqra,
> > >
> > > Thank you for your feedback! We appreciate your constructive reviews for improving our work.
> > >
> > > We admit that there are limitations that you mentioned. However, we believe that our method has more advantages than limitations regarding **enhancing the performance of knowledge-intensive tasks that require consolidating multiple pieces of evidence**. We will try to mitigate the limitations in our future revisions. Please let us know if you have any additional questions or concerns we might want to address.

---

### Author Response · Authors · 2024-11-23
**Response to all Reviewers**

We thank all the reviewers for their thoughtful and valuable feedback! We have conducted additional experiments and revised the paper based on the feedback.


## 1. Experiment with a smaller compression model with Llama3-8B-Instruct
We conducted an additional experiment by using **Llama3.2-3B-Instruct** as the compression model and Llama3-8B-Instruct  as the target model. **We have updated the results in Table 1 of our revised draft.** Despite using a smaller compression model, FaviComp with 3B compressor performs comparably with FaviComp with 8B compressor, even outperforming other baselines that use trained and larger compressors.

| Target Model: Llama3-8B-Instruct |                      |               |             |             |             |             |
| --------------------------- | -------------------- | ------------- | ----------- | ----------- | ----------- | ----------- |
|                             | Compressor           | NQ (Acc / F1) | TQA         | HotpotQA    | 2WikiMQA    | MuSiQue     |
| FaviComp                    | Llama3.2-3B-Instruct | 42.8 / 46.8   | 68.0 / 70.9 | 33.0 / 41.6 | 29.6 / 35.2 | 10.8 / 19.9 |
| FaviComp                            | Llama3-8B-Instruct   | 42.3 / 46.6   | 68.4 / 71.5 | 32.3 / 41.0 | 27.6 / 33.6 | 11.4 / 20.1 |


## 2. Head-to-Head comparison with RECOMP
We conducted a head-to-head experiment on RECOMP-abstractive by using the same base compression model as FaviComp for a more fair comparison. We construct training data on NQ, TQA, and HotpptQA according to RECOMP paper and finetune Mistral-7B-Instruct on each of the training data. **We present the evaluation settings and results in Appendix B.2 and Table 4 of the revised draft.**

|                    | Compressor                          | NQ (Acc / F1) | TQA | HotpotQA |
| ------------------ | ----------------------------------- | ------------- | -------------- | ------------------- |
| RECOMP-abstractive | Trained T5-large                    | 38.0 / 37.8   | 62.1 / 65.0    | 27.4 / 34.3         |
| RECOMP-abstractive | Trained Mistral-7B-Instruct-v0.3    | 38.3 / 38.2   | 63.0 / 65.4    | 29.5 / 36.6         |
| FaviComp           | Train-free Mistral-7B-Instruct-v0.3 | 40.3 / 40.4   | 65.9 / 68.9    | 32.0 / 40.5         |

**Even though using larger base model for compression enhances the performance of RECOMP-abstractive to some extent, it still underperforms compared to training-free FaviComp.** This underscores that the familiarization during evidence compression and integration of parametric and non-parametric knowledge are more helpful to the downstream generation than relying on a trained model for evidence compression.

---

### Note · Authors · 2024-12-14

**Comment:**

Dear Area Chair and Reviewers,

Thank you for your thorough and constructive feedback on our manuscript. After careful consideration of the reviews and feedback received, we have decided to withdraw our submission. We sincerely appreciate the time and effort the reviewers invested in providing detailed comments and suggestions. Your feedback will be valuable for improving our work.

Best regards,
The Authors

**Withdrawal Confirmation:**

I have read and agree with the venue's withdrawal policy on behalf of myself and my co-authors.